# Impact of wheat cultivar development on biomass and subsoil carbon input: A case study along an erosion-deposition gradient

Luis A. Pires Barbosa<sup>1,2</sup>\*, Martin Leue<sup>1</sup>, Marc Wehrhan<sup>1</sup>, Michael Sommer<sup>1,3</sup>

- <sup>1</sup>Research Area 1 "Landscape Functioning", Working Group: Landscape Pedology, Leibniz Centre for Agricultural Landscape Research (ZALF), Müncheberg, 15374, Germany.
  - <sup>2</sup>Research Area 4 "Simulation and Data Science", Working group: Multi-Scale Modelling, Leibniz Centre for Agricultural Landscape Research (ZALF), Müncheberg, 15374, Germany.
  - <sup>3</sup>Institute of Environmental Science & Geography, University of Potsdam, Potsdam, 14476, Germany.
- 10 \*Correspondence to: luis.pires@zalf.de

**Abstract.** Crop biomass, especially from belowground, improves soil health and recovery. However, the effects of cultivar traits and erosion on biomass production, particularly root biomass, remain unclear. We quantified root system characteristics throughout the wheat growing season, considering different cultivars and varying soil erosion states. This data informed a model assessing cultivar performance on root biomass production under different soil water and erosion scenarios. Erosion, mainly by tillage, combined to a modern cultivar reduced total wheat biomass, leaving 3.2 tons less carbon per hectare annually —of which 0.6 tons come from roots in the subsoil. The modern cultivar produced 70% more grain in depositional soils and 30% more in highly eroded soils than older cultivars. However, this increased grain yield came with a trade-off: carbon input into soils decreased by 32% in eroded soils and 43% in depositional soils. Simulations reveal that modern cultivars are more sensitive to dry soil conditions. In severely eroded areas, a 12% loss in soil water leads to a root biomass reduction of 0.05 t C ha<sup>-1</sup> in older cultivars, whereas younger varieties exhibit a much larger decline of up to 0.2 t C ha<sup>-1</sup>.

#### 1 Introduction





Long-term soil erosion exerts a major global threat to soil health and food security (Quinton et al., 2022). Beyond the significant impact of nutrient losses through lateral movement (Alewell et al., 2020; Quinton et al., 2010), erosion disrupts soil structure and alters microbial processes (Hartmann and Six, 2023). It further exerts a strong impact on the global C cycle (Bar-On et al., 2025; Doetterl et al., 2016; Van Oost et al., 2006). Tillage erosion has been identified as a very important mode of soil erosion (Borrelli et al., 2023; Wilken et al., 2020) that significantly influences the landscape-scale dynamics of carbon (C), nitrogen (N), and phosphorus (P) (Berhe et al., 2018; Juřicová et al., 2025; Nie et al., 2019) affecting crop biomass production (Kosmas et al., 2001; Öttl et al., 2021).

The drive for greater operational efficiency in crop production has led to machinery innovations that have accelerated the pace of tillage erosion, shortening its timeline from centuries (van der Meij et al., 2019; Quinton et al., 2022) to mere decades. This intensification is largely driven by increased tillage depths and speeds, which enhance soil translocation (Kietzer, 2007; Van Oost et al., 2006; Öttl et al., 2022), and hasten the leveling of hummocky landscapes (van der Meij et al., 2019; Poesen, 2018). Locations experiencing the most intense soil erosion expose subsoils enriched in mineral content characterized by unsaturated reactive surfaces, which promote the stabilization of carbon derived from plant and microbial sources (Doetterl et al., 2016; Remus et al., 2018), making them more efficient at capturing and retaining C. This potential has been modeled (Berhe et al., 2008) and experimentally confirmed for eroded (Harden et al., 1999) and depositional soils (Hoffmann et al., 2018). Besides the aspect of C sequestration in *topsoils* in disequilibrium,

C stocks might be enhanced in *subsoils* as well, where mineral surfaces remain far from carbon saturation (Georgiou et al., 2022). Therefore researchers emphasize the benefits of management practices that promote deeper carbon inputs, such as through enhanced root biomass production (Button et al., 2022).

Soil thinning on convex hilltops, steep slopes, and shoulder positions reduces local nutrient reserves and water-holding capacity (Quinton et al., 2022), ultimately leading to decreased crop biomass production (Öttl et al., 2021). In contrast, the accumulation of fertile topsoil in concave landscape positions enhances nutrient and moisture availability, supporting greater biomass production (Öttl et al., 2021). As a result, the erosion–deposition process creates spatial heterogeneity in crop growth conditions, thereby influencing food production and soil carbon inputs. The spatial heterogeneity of above-ground biomass (Herbrich et al., 2018) can be detected using the intercepted photosynthetically active radiation (IPAR) (Öttl et al., 2021), making wheat shoot a reliable indicator for mapping erosional status in hummocky landscapes. Enhancing canopy light capture through rapid early growth of leaf area or delayed leaf senescence is linked to higher yields (Parry et al., 2011). Therefore, grain yield in wheat is closely linked to above-ground biomass (Lichthardt et al., 2020). This relationship has traditionally guided breeding strategies which aimed at boosting productivity by increasing crop biomass (Sanchez-Garcia et al., 2015).

However, studies have shown that wheat cultivar development have focused solely on increasing grain yield, often at the expense of root biomass production (Rouch et al., 2023). This not only weakens crop resilience in the face of increasing climate change challenges (Martre et al., 2024; Qiao et al., 2022), but also jeopardizes the recovery of eroded soils and neglects their potential for enhanced C sequestration. It is known that bigger root system may be associated with enhanced water and nutrient uptake, leading to higher grain yields (Cormier et al., 2016). Root traits like depth and density are particularly critical for N capture (Holz et al., 2024), further supporting yield improvements (Martre et al., 2024). Moreover, root biomass is vital for maintaining and increasing soil organic carbon (SOC) because it remains in the soil two to three times longer than C from other above ground biomass or manure (Kätterer et al., 2011). This makes it a more durable and effective contributor to long-term soil health. Given the potential of root biomass to enhance C sequestration, wheat cultivar development aiming at increased grain yield and root biomass offers a promising management option (Heinemann et al., 2023; Rouch et al., 2023). Nonetheless, a comprehensive understanding of how soil erosion affects carbon inputs from crop root biomass remains a significant gap in current research. Although such understanding is essential for scaling subsoil carbon sequestration potential to larger spatial scales. Field experiments evaluating above-and belowground biomass production of different wheat cultivars across the full erosion–deposition gradient are still lacking.

Although cultivar traits plays a significant role in root system development, root distribution is heavily influenced by soil characteristics, nutrient availability (Robinson, 2001), and mechanical impedance (Keller et al., 2019). Estimating root development is inherently challenging and requires specialized techniques for accurate quantification. Traditionally, destructive methods such as soil coring, whole root system excavation, and trenching have been employed (Milchunas, 2009), but these approaches heavily disturb the experimental plot and offer only a single time-point snapshot of the root system. In contrast, nondestructive methods like minirhizotrons provide an in situ approach for observing root growth over time. Minirhizotrons allow for the simultaneous measurement of root production and disappearance (Johnson et al., 2001). Root turnover during crop development has been shown to significantly impact C balance (Milchunas, 2009; Remus and Augustin, 2016) and is fundamental for quantifying the transfer of recently assimilated carbon from shoots to the root–soil system (Swinnen et al., 1994). This dynamic measurement combined to data-driven machine learning technique may provide an opportunity to explore a potential correlation between intercepted photosynthetically active radiation (IPAR) and root biomass production, as both can be quantified simultaneously. Such a correlation may be specific to the soil type in which a given cultivar develops, reflecting interactions between soil properties and cultivar

performance. To our knowledge there is no study that track root biomass dynamics throughout the crop season describing a correlation with aboveground measure.

We hypothesize that the dynamics of root biomass throughout the crop season are shaped by the interaction between wheat cultivar traits and landscape erosion–deposition gradient—whereby older cultivars, with their deeper and more extensive root systems, may offer potential to boost subsoil carbon inputs in degraded soils. This interplay is expected to influence also aboveground biomass. To test this hypothesis, we selected wheat cultivars representing a century of regional breeding history. We quantified root system development throughout a complete growing season along a full erosion–deposition gradient. This enabled an assessment of the relationship between root biomass and intercepted photosynthetically active radiation (IPAR) along the erosion–deposition gradient under varying soil moisture levels.

#### 2 Material and methods







## 2.1 Experimental configuration

A plot experiment was set up at the CarboZALF field site (53°23' N, 13°47 E) located in the hummocky "Uckermark" region of North-Eastern Germany (Sommer et al., 2016). Our field site represents a typical, erosion-affected soil landscape evolved from glacial till (Sommer et al., 2008) in which tillage erosion had been shown to be the major mode of soil erosion (field scale: Wilken et al. (2020); landscape scale: Öttl et al. (2021)). We selected four soils along the full gradient of erosion-deposition (Fig.1): (i) An extremely eroded soil at a steep slope (Calcaric Regosol, Rg-ca), (ii) a less eroded soil at a flatter midslope (Nudiargic Luvisol, Lv-ng), (iii) a non-eroded soil at plateau position (Calcic Luvisol, Lv-cc), and, (iv) a depositional soil (Gleyic-Colluvic Regosol, Rg-co.gl) (IUSS, 2015) located at the fringe of a topographic depression (kettle hole). Here, groundwater influence is documented by redoximorphic features in the upper 1m. Because of lateral soil translocation the selected sequence represents a large gradient in crop growth conditions as well, especially for water and nutrient supply (for details, see Table S1 in the supplement).

Figure. 1: Soils along the erosion-deposition gradient (soil classification according to WRB, IUSS 2015); for soil properties see Table S1 (supplement).

Bulk density was measured using undisturbed soil cores of 100 cm<sup>3</sup>. Bulk soil samples were air dried, gently crushed and sieved at 2mm to separate the fine earth fraction (< 2 mm) from the gravel (> 2 mm). The particle size distribution of the fine earth was determined by a combined wet sieving (> 63  $\mu$ m) and pipette (< 20  $\mu$ m) method; pretreatment for particle size analysis was performed by wet oxidation of the OM using H<sub>2</sub>O<sub>2</sub> (10 Vol. %) at 80 °C and

dispersion by shaking the sample end over end for 16 h with a 0.01 M Na<sub>4</sub>P<sub>2</sub>O<sub>7</sub> solution (Schlichting et al., 1995). Soil pH was measured using a glass electrode in 0.01 M CaCl<sub>2</sub> suspensions at a soil to solution ratio of 1:5 (w/v) after a 60-min equilibration period. Total C and N was determined by dry combustion using an elemental analyzer (Vario EL, Elementar Analysensysteme, Hanau, Germany). CaCO<sub>3</sub> was determined conductometrically using a Scheibler apparatus (Schlichting et al. 1995). The SOC content was calculated as the difference between total C and CO<sub>3</sub>-C.

To test our hypothesis we set up a randomized block design with four soil types and three cultivars of winter wheat in four replicates (Fig.2). To cover the centennial breeding progress in the Uckermark region the following varieties were selected: Ostpreussischer Eppweizen (1910), Hadmerslebener Qualitas (1957), and Ponticus (2015). By this selection we want to check, whether root traits of cultivars changed over decades like grain:straw ratios did and, therefore, modulate long-term C inputs into subsoils. At each block wheat cultivars were sown in each four adjacent plots with parallel stripes of 1.5 m width and 7 m length (Fig.2). Sowing date was on October, 6th, 2022, (300 grains per m², seed germination October, 17th, preceeding crop: summer barley). Harvest took place on July, 27th, 2023. Each plot was fertilized with 80 kg N ha-1 (calcium ammonium nitrate, May, 5th, 2023), 27 kg Mg ha-1 and 22 kg S ha-1 (February, 28th, 2023). Annual precipitation in 2022 and 2023 and annual mean temperatures in 2022 and 2023 are provided in the supplement (Figure S1).

Figure 2: Image of crop canopy on June 21st, 2023 (left); single block design (right); O = Ostpreußischer Eppweizen (1910); H = Hadmerslebener Qualitas (1957), P = Ponticus (2015).

### 2.2 Intercepted Photosynthetically Active Radiation (IPAR)

The canopy analysis system sensor SunScan (SunScan SS1; Delta-T Devices Ltd., Cambridge, UK) combined with its BF5 sensor, capable of measuring incident and transmitted Photosynthetically Active Radiation (PAR), was used to calculate Intercepted Photosynthetically Active Radiation (IPAR) and Leaf Area Index (LAI). These measures were carried out at BBCH 37, 69, 85 and 92.

# 2.3 Above-ground biomass







Above-ground biomasses (total shoot, straw and grain) were determined gravimetrically from wheat plants harvested from 1 m<sup>2</sup> subplots in four replicates for BBCH 92. Dry mass was measured after drying samples at 60°C for over 24 hours (Herbrich et al., 2018).

## 2.4 Rhizotron installation and Rhizoscanning

Immediately after the seeding, a soil core of 1.55 m length and 6.3 mm in diameter was drilled at the center of each wheat stripe using a tractor-driven apparatus (Leue et al., 2019). A transparent acrylic glass tube of 1.6 m length and 6.2 mm outer diameter was installed in each drill hole. In sum over four soil sites, four plots per sites, and three wheat cultivars per plot, 48 tubes were installed. The above-ground part of the tubes was covered with black tape in order to minimize light and moisture intrusion and heat exchange (Herbrich et al., 2018).

At 12 dates (BBCH-scale) plus one after harvest (Fig. 3), rhizoscans were taken inside the tubes using the minirhizotron imaging system CI 600 (Root Imager CI 600; CID Inc., Camas, USA): October 26<sup>th</sup> 2022 (BBCH Rg-ca: 11-12, other sites: 12), November 07<sup>th</sup> (BBCH RG-ca: 13, other sites: 13-15), November 21<sup>st</sup> (BBCH Rg-ca: 13-15, other sites: 22), January 10<sup>th</sup> 2023 (BBCH Rg-ca: 20-23, other sites: 21-26), February 06<sup>th</sup> (BBCH Rg-ca: 20-23, other sites: 21-26), March 21<sup>st</sup> (BBCH 24-29), April 4<sup>th</sup> (BBCH 24-29), April 26<sup>th</sup> (BBCH Rg-ca: 30, other sites: 31), May 22<sup>nd</sup> (BBCH 37), June 14<sup>th</sup> (BBCH 69), July 4<sup>th</sup> (BBCH 85) and July 27<sup>th</sup> 2023 (BBCH 92).

The rotating scanner yields 360°-images of the tube-soil interface with 215.9 mm height and 195.7 mm width at a resolution of 300 dpi. After each scan, the scanner is pushed perpendicularly downwards in fixed steps. A horizontal overlapping of 1 cm ensured the later merging of the images. Particular attention was payed to maintaining both the soil and the plant population around the tube undisturbed.

# 2.5 Root segmentation and image analysis









The raw images from rhizoscan underwent processing in two steps. Firstly, a deep-learning model (Smith et al., 2022) was trained on a dataset comprising varied root shapes, colors, and soil types (background). The training phase involved annotating a region of interest (ROI 700x700 pixels) from 257 raw images using Root Painter (Smith et al., 2022). This process entailed manually selecting roots and distinguishing them from soil or other non-root objects. The root segmentation model was evaluated by comparing the annotated and trained dataset with the predicted results. Accuracy and precision were then measured for each annotated image. Subsequently, the trained model was utilized to segment all 560 images in the dataset. Secondly, the segmented images, disregarding the first 20 cm, underwent filtering and standardization to eliminate non-root objects (such as soil particles, water drops, and straw), after which the root volume and average diameter were quantified using RhizoVisionExplorer (Seethepalli et al., 2021). This involved considering the average diameter and overall length for cylindrical roots. The model used for this considered broken roots, since a continuous image was not always possible and the pixel threshold for non-roots objects as well as root pruning were set to 5. Based on the measured root diameter [mm] and a constant specific root length of 100 mg<sup>-1</sup> (Herbrich et al., 2018), root tissue density (RTD [g mm<sup>-3</sup>]) was calculated (Rose, 2017). By combining the measured root volume (mm<sup>3</sup>) with RTD, we determined the root mass [g] across all crop development stages. Finally, using root mass per plant and the number of plants harvested per square meter, we converted the root mass to root biomass (RB [g m<sup>-2</sup>]) for each crop development stage.

For grain, straw and roots we used a carbon content of 45% to convert the respective biomasses into carbon masses (Bolinder et al., 2007). Root exudates was not quantified in our field experiment, similar to the approach taken by Heinemann et al. (2023). In essence, our calculated belowground root biomass represents the minimum C input into subsoils.

#### 2.6 Unsupervised and supervised machine learning (ML)

Principal component analysis (PCA) was performed via Scikit-Learn package in Python (Pedregosa et al., 2011). The clustering of data, variable loadings and their relationships were evaluated. Two principal components were selected and the total variance explained by each component (soil density, pH, CaCO<sub>3</sub>, root diameter, SOC, N, LAI, IPAR, root

depth and available water content (AWC)) was quantified. Based on this exploration, we defined a set of variables to perform the yield prediction by random forest resemblance algorithm.

Data-driven machine learning technique was employed to analyze the importance of each feature on the output. Therefore, the independent variables (features) were defined as IPAR, soil water, soil plot and wheat cultivar to explain the output measured by root biomass (dependent variable). A total of 30% of the samples were used to train the multiple output regression random forest ensemble learning algorithm (Pedregosa et al., 2011). The trained algorithm was used to predict new data combining variables and output. For the assessment of the regression algorithm (Geron, 2017) the metrics were: 1) coefficient of determination R2-score, which is the proportion of the variance in the output that is predictable from the variable (the best possible score is 1.0) and 2) Mean Absolute Percentage Error (MAPE), which finds all absolute errors (xi - x), adds them all and divide by the number of errors. In random forest regressor, the depth of 100 was used as a decision node in a tree can be used to assess the relative importance of that variable in predicting the output with 1000 estimators. For this calculation, whose values are positive and sum to 1.0, the higher the value, the more important is the contribution of the variable to the prediction function. The mean decrease impurity method available in scikit-learn was applied for each output.

The trained algorithm was employed to estimate the relationship between IPAR (Intercepted Photosynthetically Active Radiation) and root biomass production across various soil erosion gradients (Rg-ca, Lv-ng, Lv-cc, and Rg-co.gl) and soil water content levels (18, 21, 24, 27 and 30 cm<sup>3</sup> cm<sup>-3</sup>).

#### 2.7 Statistical analyses

All statistical analyses were carried out in the R software package (R Core Team, 2017). Analyses of differences among treatments were performed using a two-way analysis of variance (ANOVA). Differences between data sets were considered significant at p < 0.01 and p < 0.05 for the feature importance. Tukey's post hoc test was used.

## 3 Results







## 210 3.1 Shoot biomass

The total shoot biomass decreases with the increase in soil erosion degree (Figure S2), with wheat in Rg-ca at knolls producing 500 g m<sup>-2</sup> less biomass in comparison to Rg-co.gl at depressions. The more recently developed cultivar (Ponticus) tend to produce greater shoot biomass than the other cultivars of the study, but without significant differences within soil types. When analysing grain production (Fig. 3a), it is directly proportional to the age of the wheat cultivars, with a more pronounced effect in soils with lower erosion levels. The earlier developed cultivar (Ostpreußischer) in Rg-ca produced 44% less grain than the Ponticus cultivar in Rg-co.gl. Straw production is also directly proportional to the degree of soil erosion but inversely proportional to the age of the wheat cultivar (Fig. 3b). In this case, the Ponticus cultivar produced 50% less straw mass than the Ostpreußischer cultivar when comparing the extremes along the erosion-deposition gradient (Rg-ca and Rg-co.gl). This ratio between grain and straw biomass resulted in a nearly constant harvest index (HI) (Figure S3) along the erosion-deposition gradient for the more recent wheat cultivars: Ponticus had values close to 0.7, and Hadmerslebener had values around 0.6. However, the earlier developed cultivar (Ostpreußischer) showed a greater dependency of the HI on the degree of soil erosion by a HI reduction of 45% from the most eroded soil (Rg-ca) to the deposition soil (Rg-co.gl).

Figure 3. Crop biomass [g m<sup>-2</sup>] and corresponding carbon (C) masses [g m<sup>-2</sup>] for the three wheat cultivars along the erosion-deposition gradient. (a) Grain yield, (b) Straw biomass, and (c) Root biomass measured at growth stage BBCH 69. The letters above the boxplots indicate significant differences in grain yield between treatments (p < 0.05), based on post-hoc tests.

## 230 3.2 Root biomass







At BBCH 69 no significant differences of root biomass were observed among cultivars within the same soil type. However, cultivars grown on Regosols produced approximately 85 g C m<sup>-2</sup>, while those on Luvisols produced around 175 g C m<sup>-2</sup> (Fig. 3c). The dynamics of root growth reveal a greater root cross-sectional area in Luvisols compared to Regosols (Fig. 4). Specifically, the initial development of the root for all wheat cultivars was hindered in the Calcaric Regosol (Rg-ca) due to the very dense glacial till (C horizon) starting at 25 cm depth (Table S1 and Fig. 4). At the Rg-ca, root biomass was highest around BBCH 85, with Ostpreußischer showing slightly higher values (200 g m<sup>-2</sup>) than the other cultivars (Figure S4). In Nudiargic Luvisol (Lv-ng) soil, all cultivars exhibited higher root biomass around BBCH 69 (Fig. 3c, Figure S4), with Ostpreußischer leading with approximately 400 g m<sup>-2</sup>. In Calcic Luvisol (Lv-cc), all cultivars exhibited higher root biomass around BBCH 69 (Fig. 3c, Figure S4), with Hadmerslebener leading (550 g m<sup>-2</sup>), then declining sharply after BBCH 85. In Gleyic-Colluvic Regosol (Rg-co.gl), root biomass is more evenly distributed between BBCH 37 and 92, with Ostpreußischer again showing higher values overall (200 g m<sup>-2</sup>). Between BBCH 37 and 92 is the moment when the most root branches are formed (Figure S5). For the BBCH 69, wheat cultivars Ostpreußsicher and Ponticus, an increasing trend in the maximum total root biomass was observed for the sequence of soils: Rg-ca < Rgco.gl < LV-cc < LV-ng. For the Hadmerslebener wheat the order was: Rg-ca < Rg-co.gl < LV-ng < LV-cc. The root volume concentration, calculated as the integral of root area along depth (Fig. 4) for BBCH 69, exceeds 50% below 1 meter in Luvisols. In Regosols, this value is less than 50%. However, in Rg-ca, younger varieties concentrate only 15% of their roots below 1 meter, while the oldest variety exceeds 40%.

Luvisols exhibited the highest root biomass growth rate (approximately 10 g m<sup>-2</sup> day<sup>-1</sup>) (Figure S6), while the highest rates in Regosols are around 5 g m<sup>-2</sup> day<sup>-1</sup>. For all soil types, the peak of root growth rate occurred at 228 days after sowing (BBCH 37), except for Rg-ca, where it occurs at 251 days (BBCH 69). Ostpreußischer and Ponticus showed the highest values in Rg-ca and Lv-ng, whereas in Lv-cc and Rg-co.gl, Hadmerslebener and Ostpreußischer, respectively, outperformed the other cultivar. The maximum decomposition rate (negative growth rate) occurred around 293-350 days (after BBCH 92) for all soils and wheat cultivars. The highest degradation rate (around 5 g m<sup>-2</sup> day<sup>-1</sup>) was observed in Lv-ng and Lv-cc for the Ponticus and Ostpreußischer cultivars, respectively. The greatest root concentration was observed at BBCH 37 and 69 (228 and 251 days after sowing) of the Hadmerslebener at Lv-cc between the soil layers of 0.8 and 1.2m depth followed by Ostpreußischer at Lv-ng at depth between 0.8 and 1.6m (Fig. 3). At Rg-ca, Ostpreußischer presented the greatest homogeneity of root distribution profile along soil depth. The highest root degradation appeared in Rg-ca for the wheat cultivars Ponticus and Ostpreußischer at soil depth of 0.8 and 1.2 m respectively.

Figure 4: Representation of the soil profile for each plot, accompanied by average root area values measured across soil depths for each wheat cultivar over time.

#### 3.3 Root to shoot ratio








No clear trend can be observed in the root-to-shoot ratio across soil erosion levels and wheat cultivars (Figure S7). The values for Ostpreußischer fluctuated around 0.28. However, Hadmerslebener presented the lowest value in Rgca and the highest value in Lv-cc, with a difference of more than four times. Ponticus presented the highest value in Lv-ng and the lowest value in Rg-co.gl, with a difference of more than two times.

## 3.4 Modeling root biomass production across erosion levels

Principal component 1 (PC1) explains 53.7% of the total variance in the dataset while PC2 explains 16.2%, a total of 69.9% of the variance (Fig. 5a). Colored confidence ellipses representing the clustering of soil types by erosion-deposition gradient indicating that variables like density, pH and Calcium Carbonate (CaCO<sub>3</sub>) had a stronger negative effect on Rg-ca while SOC and N had stronger and positive effect on Rg-co.gl. Soils with higher available water capacity or/and capillary rise of ground water (e.g. Rg-co.gl) produces shallower roots while denser subsoils (e.g. Rg-ca) produces larger-diameter roots (Fig. 5a).

The supervised machine learning regression achieved a coefficient of determination (R²) of 0.89, with a Mean Absolute Percentage Error (MAPE) of 26% (Figure S8). According to the random forest ensemble algorithm the IPAR was the most important factor for predicting the root biomass values (Fig. 5a). All wheat cultivars in all soil types exhibited a similar asymptotic regression model for the relationship between root biomass and IPAR (Fig. 6). This relationship featured an initial rapid increase in biomass with increasing IPAR, followed by a plateau, indicating a saturation point where additional light absorption no longer significantly boosted root biomass. The plateau values varied based on soil water content, wheat cultivar, and soil erosion degree, as highlighted by the order of importance in Fig. 5b.

At a model-assigned soil water content of 18 cm³ cm⁻³, root biomass plateaued at lower levels compared to higher water contents. Among the cultivars, Ostpreußischer generally exhibited the highest root biomass across all soil types at this water content. At 24 cm³ cm⁻³, the shape of the graphs and the plateau values of root biomass remained consistent with those observed at the lower water content. At 30 cm³ cm⁻³, the highest biomass values were recorded across all soil types, showing a more rapid increase in biomass with IPAR. Hadmerslebener outperformed most of the other cultivars under these conditions. Luvisols (Lv-ng and Lv-cc) consistently supported higher root biomass compared to Regosols (Rg-ca and Rg-co.gl) across all wheat cultivars and soil water contents, with differences reaching up to 50% at higher water contents. The maximum observed biomass was 450 g m⁻² for Lv-cc soil with a water content of 30 cm³ cm⁻³, achieved by Hadmerslebener. In contrast, the minimum observed value was 180 g m⁻² for Ponticus in Rg-ca soil with a water content of 18 cm³ cm⁻³.

Simulations revealed that the two more recent wheat cultivars experienced an average root biomass reduction of 4 to 8 g m<sup>-2</sup> per 1% decrease in volume of soil water content, showing greater sensitivity compared to the 2 g m<sup>-2</sup> reduction observed in the earlier developed cultivar (Table S2). In severely eroded areas, a 12% loss in soil water leads to a root biomass reduction of 0.05 t C ha<sup>-1</sup> in older cultivars, whereas younger varieties exhibit a much larger decline of up to 0.2 t C ha<sup>-1</sup>. In depositional soils these values increased to 0.12 and 0.23 t C ha<sup>-1</sup>, respectively.

Figure 5: Unsupervised and supervised machine learning. a) Visual representation of the principal component analysis (PCA), showing the relationships between different variables across four groups of soils (Rg-ca, Lv-ng, Lv-cc, and Rg-co.gl). Confidence ellipses: Represent the clustering of different soil types based on the PCA. b) Importance of each feature in the prediction of root biomass.

Figure 6: Simulated relationship between root biomass (g m<sup>-2</sup>) and IPAR (incident photosynthetically active radiation) along the erosion-deposition gradient and the three wheat cultivars. Each soil type is represented in a separate row, while each column corresponds to a different level of soil water content (18, 24, and 30 cm<sup>-3</sup>).

# 4 Discussion



Along the studied erosion–deposition gradient, the oldest wheat cultivar grown in Rg-co.gl soil produced approximately 50% more straw and root biomass than the most recent cultivar grown in Rg-ca soil—corresponding to an annual difference of about 3.2 tons of C per hectare—of which 0.6 tons come from roots in the subsoil (Fig. 3). This difference underscores the importance of wheat cultivar selection, as biomass production and allocation strategies vary among cultivars. Earlier developed cultivars exhibit significant higher straw biomass production than the more recently developed cultivar, while recently developed cultivars face greater declines in grain yield under eroded soil conditions. Earlier-developed wheat cultivars demonstrate greater resilience in root biomass production, even at the extremely eroded soil with greater root development at depths beyond 1 meter compared to the other cultivars (Fig. 4). Given the benefits

of promoting deeper carbon inputs within the soil profile (Georgiou et al., 2022) such performance of Ostpreußischer cultivar is a desirable trait for increasing carbon stocks in degraded soils (Button et al., 2022).









The comparison of root biomasses with those reported in the literature presents several challenges, primarily due to the wide variation in sampling methods and data processing (Milchunas, 2009). Among these factors, the sampling date stands out as particularly critical. Our results reveal that root biomass formation and degradation is highly dynamic throughout the crop cycle, especially between BBCH stages 12 and 37, with a sharp decline following harvest (Figure S4). Heinemann et al. (2025), using soil core sampling one week after harvest, reported values ranging from 86 to 267 g m<sup>-2</sup> — comparable to those obtained in our study for the same period. It is important to acknowledge that, despite efforts to minimize it, minirhizotron-based measurements inherently carry a degree of error that may lead to the underestimation of root biomass (Johnson et al., 2001). To improve measurement accuracy, we excluded the top 20 cm of soil, where nonroot materials such as crop residues and straw can interfere with image analysis. While this approach reduced potential noise in the data, it also resulted in the exclusion of shallow roots, thereby contributing to a likely underestimation of total root biomass. Hirte et al. (2018), also using soil cores, reported values between 87 and 274 g m<sup>-2</sup> for the BBCH 37 to 69 stage range, whereas our values for the same period ranged from 48 to 491 g m<sup>-2</sup>. Another important aspect is the sampling depth: while the comparative studies reached a maximum of 1 meter, our data reveal substantial root biomass between 1.0 and 1.6 meters deep (Fig. 4). In a study conducted in the same experimental area, Herbrich et al. (2018), using the same sampling method, reported root biomass values at BBCH 71 of  $76 \pm 9\,\mathrm{g\,m^{-2}}$  in the most eroded soil and 246 ± 18 g m<sup>-2</sup> in the depositional soil. These values align well with our findings at BBCH 69, which ranged from  $99 \pm 12$  g m<sup>-2</sup> in eroded soils to  $215 \pm 26$  g m<sup>-2</sup> in depositional areas.

Shoot biomass was similar across the wheat cultivars and was only significantly reduced in the most eroded soil (Figure S2). This finding is consistent with previous studies, where tillage erosion plays a key role in driving spatial variability in biomass production, especially in relatively dry arable hummocky landscapes (Öttl et al., 2021). This is evident from the lowest Leaf Area Index values (used as a proxy for above-ground biomass), which are often found on hilltops, where tillage erosion is most severe (Kosmas et al., 2001). Although the study by Kosmas (2001) was conducted in a different area, the absolute values of wheat biomass observed ranging from 100 to 1000 g m<sup>-2</sup> are slightly lower than our results (Figure S2), ranging from the most eroded soil to the soil formed by surface deposition, respectively. The general pattern of lower biomass on eroded hilltops compared to depositional areas is observed across various common crops in Quillow catchment (Öttl et al., 2021). When splitting shoot biomass into grain and straw, the spatial variability is even greater than in total shoot biomass, due to the varying responses of each wheat cultivar to erosion-related soil properties (c.f. Fig. 3). Wheat cultivars show different strategies of biomass allocation and resilience to soil erosion. For instance, the two younger cultivars maintain a steady harvest index (HI) as both grain and straw biomass increase proportionally with decreasing erosion (Figure S3). In contrast, the oldest cultivar produces more straw than grain as erosion lessens (Fig. 3), leading to a lower harvest index (Figure S3).

In deposition soil, recently developed wheat cultivars produce nearly 70% more grain compared to earlier developed cultivars in deposition soil. In highly eroded soils, recently developed wheat cultivars produce nearly 30% more grain yield compared to earlier developed cultivars (Fig. 3). However, the increase in grain yield comes with a trade-off: there is 32% less C input (from straw and root) into highly eroded soils and 43% less C input into depositional soils (Fig. 3). In agricultural soils, crop roots represent a major source of soil organic carbon (SOC) (Poeplau et al., 2021). Root-derived carbon is particularly important because it persists in the soil two to three times longer than carbon from above-ground residues or manure (Kätterer et al., 2011), making it a crucial contributor to SOC maintenance and accumulation. In our study, root carbon inputs ranged from 27 to 134 g C m<sup>-2</sup> in Luvisols and from 6 to 51 g C m<sup>-2</sup> in Regosols (Figure

S4, after harvest). These values slightly deviate from the values 47–63 g C m<sup>-2</sup> reported by Hirte et al. (2018), who sampled to a depth of 75 cm also after harvest. As previously discussed, our measurements extended beyond 1 m depth, where over 50% of the root volume is concentrated in Luvisols (Fig. 4), likely explaining the higher values observed.

Using carbon partitioning Swinnen et al. (1994) reported a total shoot growth of 5.7 t C ha<sup>-1</sup> over the growing season of spring wheat, with 0.9 t C ha<sup>-1</sup> allocated to root biomass. These values closely match the averages observed for cultivars grown in Rg-co.gl soil, which produced 5.7 t C ha<sup>-1</sup> aboveground and approximately 1 t C ha<sup>-1</sup> in root for BBCH 69 (Fig. 3). Additionally, the authors found that root respiration and rhizodeposition contributed an additional 0.9 and 0.5 t C ha<sup>-1</sup>, respectively, for the same level of shoot production.









Recognizing the importance of root respiration and rhizodeposition during plant development, Remus and Augustin (2016) demonstrated that belowground carbon transfer is highly dynamic during crop season. This is agreement with root biomass dynamics measured with minirhizotrons throughout the crop season (Fig. 4 and Figure S6). In both Luvisols, the root system reaches maximum depth around the BBCH 29 growth stage (Fig. 4), whereas, in Regosols, root penetration continues up to the BBCH 37 and 69 stages in the more eroded and depositional soils, respectively (Fig. 4). This delayed root development in highly eroded soils contributes to reduced productivity, underscoring the challenges of managing such degraded landscapes for agricultural use. The transfer of recently assimilated C from the shoot to the rootsoil system varies with different stages of crop development was confirmed by the root biomass rate with the maximum C transfer from the shoot to the root system during the start of stem elongation (BBCH 37). The effect was regardless of soil erosion status (Figure S6) as also stated by previous work (Hoffmann et al., 2018). Stem elongation is the moment when the most root branches are formed (Figure S5) increasing significantly the root area (Fig. 4). Higher soil densities slow root elongation rate and consequently root biomass rate (Figure S6), requiring more time for roots to penetrate deeper layers (Keller et al., 2019). The presence of dense glacial till near surface typically acts as a limiting factor for root development (Stock et al., 2007), leading to reduced root subsoil penetration in the highly eroded soil (Fig. 4). This soil displays a shallow C horizon with high soil density (Table S1) that restricts root penetration and increase root diameter (Fig. 5a and Figure S5) in Rg-ca, in agreement with previous observations (Popova et al., 2016). Root depth is also reduced in Rg-co.gl (Figure S5) despite higher SOC and N; however, this is likely due to standing groundwater at 1m, hence O<sub>2</sub> deficiency for crop roots below that depth.

Such dynamic measurements of root biomass has a correlation with aboveground biomass, using IPAR as a proxy. Based on these data, our trained machine learning algorithm suggests a logarithmic relationship; however, this correlation cannot be generalized across all wheat cultivars and soil erosion conditions. Each cultivar has distinct parameters that are further influenced by erosion and water content, making the interaction between IPAR, biomass production, and grain yield highly context-specific (c.f. Fig. 6). The simulations across erosion-deposition gradient (Table S2) highlight a concerning trend: root biomass of recently developed wheat cultivars exhibit greater sensitivity to declining soil water levels compared to earlier cultivars. This suggests that advances in cultivar development have not significantly improved climate resilience, particularly for root biomass production. With soil water levels decreasing globally (Qin et al., 2023), the capacity of the more recent wheat cultivar to sustain root carbon inputs in eroded soils appears limited. Beyond the potential for gross C input, restoring soil structural health is critical for reviving its environmental functions. In heavily eroded Rg-ca, where the structure is degraded (Barbosa et al., 2024), root system architecture can enhance soil recovery by promoting hierarchical aggregation at both micro and macro levels (Poirier et al., 2018). Root-induced biopores enrich microbial biomass within the drillosphere, which serves as a biological "hot spot" (Bundt et al., 2001). Elevated microbial activity (Leue et al., 2021) and a higher proportion of OC in these areas promote the production of extracellular polymeric substances (Carrel et al., 2017), significantly affecting water and solute transport within the soil, playing a pivotal role in soil function. The historical development of cultivars with higher root branching and expanded root area throughout the soil profile has been particularly beneficial for optimizing soil water exploration (Aguirrezabal et al., 1993). This root system robustness helps maintain biomass production even under conditions of soil hydric stress (Fig. 6), highlighting the importance of varietal development to achieve higher grain yields based not only in photosynthetic capacity and efficiency but also on crop biomass (Lichthardt et al., 2020), preferably root biomass, to assist soil structural recovery and improve crop resilience and water use efficiency (Qiao et al., 2022). This falls within the context of sustainable agricultural practices (Piñeiro et al., 2020) (e.g. mulching, reduced or no-tillage, and crop rotation) enhancing the benefits of robust root systems by minimizing soil structural disturbance, and preserving SOC.

We recommend that future research go beyond the scope of this case study, which was limited to a single field and three wheat cultivars. Expanding the study to include larger sample sizes and experimental areas across different regions with erosion—deposition gradients would provide a broader understanding of cultivar performance. This would allow for more robust conclusions and help assess the generalizability of the observed patterns across diverse environmental and management conditions. Additionally, future studies should aim for a deeper exploration of the internal mechanisms, incorporating causal analyses to better uncover the underlying biological processes and ecological dynamics that drive cultivar responses to soil redistribution.

## **5 Conclusions**









Root biomass formation and decay are highly dynamic processes throughout the crop season, shaped by both environmental and genetic factors. Soil erosion has a clear negative impact on root development, while earlier-developed cultivars positively influence root biomass—particularly at depths beyond 1 meter. Along the studied erosion—deposition gradient, the oldest wheat cultivar grown in Rg-co.gl soil produced approximately 50% more straw and root biomass than the most recent cultivar in Rg-ca soil. This corresponds to an annual difference of about 3.2 tons of carbon per hectare, of which 0.6 tons originated from subsoil roots.

Although recently developed cultivars deliver higher grain yields—up to 70% more in depositional soils and 30% more in highly eroded soils compared to older cultivars—this gain comes at a cost. The increase in yield is accompanied by a reduction in carbon inputs: 32% less in highly eroded soils and 43% less in depositional soils, due to lower contributions from straw and root biomass. In our study, root carbon inputs into subsoils ranged from 27 to  $134 \text{ g C m}^{-2}$  in Luvisols and from 6 to  $51 \text{ g C m}^{-2}$  in Regosols.

The quantification of above- and belowground biomass revealed a logarithmic relationship between root biomass and intercepted photosynthetically active radiation (IPAR); however, this correlation is highly context-dependent, varying with cultivar type and soil conditions. Simulation results further show that modern cultivars are more sensitive to dry soil, in severely eroded areas, a 12% loss in soil water leads to a root biomass reduction of 0.05 t C ha<sup>-1</sup> in older cultivars, whereas younger varieties exhibit a much larger decline of up to 0.2 t C ha<sup>-1</sup>.

To better understand and generalize these findings, long-term experiments are needed across diverse regions and erosion—deposition gradients. Such studies would enhance our understanding of cultivar performance under water-limited conditions and support more informed breeding and land management strategies that balance yield optimization with soil carbon sustainability.

#### Acknowledgements

This work was supported by the Deutsche Forschungsgemeinschaft (DFG), under Project Number 422576233 (SO 302/12-1), as part of the study "Tillage erosion affects crop yields and carbon balance in hummocky landscapes," led by Professor Dr. Michael Sommer and Professor Dr. Peter Fiener (U Augsburg). We extend our gratitude to Natalie Papke, Lidia Völker, and Sylvia Koszinski for their invaluable assistance with root measurements. We also thank the team at the

Dedelow experimental infrastructure platform for their dedicated efforts in agricultural management, harvesting, and providing above-ground plant data.

## **Author contributions**

M.S. conceived of the presented idea. L.A.P.B. structured and took the lead in writing the manuscript. M.S., M.L., and M.W. conceived and planned the experiments. M.L. and M.W. carried out the experiments. L.A.P.B. performed all image analyses, planned and executed the simulations and data analysis. M.S., M.L., M.W., and L.A.P.B. contributed to the interpretation of the results. All authors provided critical feedback and helped shape the research, analysis, and manuscript.

# 450 Competing interests

The authors declare no competing interests.

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
