# Peer review of "Impact of wheat cultivar development on biomass and subsoil carbon input: A case study along an erosion-deposition gradient"

_EGUsphere, 2025_

## Author Response (AR1)

L165ff: This section describes supervised and unsupervised ML methods. Various soil parameters are used for PCA, such as BD, C, N, P, pH, CaCO3 and others. The methods section does not describe the analysis of these parameters, nor does it provide references to other publications where these parameters have been analysed. These data are also presented in the Supplementary Table S1 for the different soil types and also for the horizons. However, it remains unclear where the data comes from or how they were measured.

Authors' response: We appreciate this observation. This information is now presented starting at line 105 of the revised manuscript.

When comparing the values for the different horizons, it is very remarkable that they are similar, sometimes down to the decimal place. From my experience with soil analysis I am not familiar with such results. These parameters are not only used for the ML, but also used for the discussion, as they have a significant influence on the work presented. Since a significant part of the work presented is based on these values, the analysis of it should be presented in a comprehensible way.

Figure 2: please adapt the figure, as it is hard to read the content (too small). Please indicate what the black dots represent (outlier or means? as the bars are quite small, it is also hard to distinguish between dots or letters)

Authors' response: The values presented in Table S1 have been thoroughly revised, and the PCA analysis has been reformulated accordingly. Some of the similar values (identical to the decimal place) result from the relatively low variability of the corresponding soil properties and the precision limits of the analytical methods used as described in line 105. Nevertheless, all data were collected at the central laboratory of ZALF, using reliable instrumentation, standardized procedures as well as internal standards.

All figures have also been revised and are now presented in an updated format.

This MS focuses on the research regarding the impact of wheat cultivar development on biomass production and carbon input in tillage-eroded soils. Through experiments setting different wheat cultivars and soil erosion gradients, it is found that soil erosion can affect wheat biomass and carbon input, and both newly developed and earlier cultivars have their own advantages and disadvantages in this regard. This conclusion provides a reference for cultivar selection and the sustainable development of agriculture. However, there are still some issues in this article:

1. When expounding on the research background, the writing logic is rather jumpy. It directly transitions from the global soil organic carbon content and the impact of tillage erosion on the carbon cycle to the acceleration of tillage erosion by agricultural machinery innovation, lacking necessary connections. For example, between "Annually, tillage erosion displaces around 0.5 Pg of this carbon pool (Quinton et al., 2010). Beyond the significant impact of nutrient loss through lateral movement (Alewell et al., 2020; Quinton et al., 2010), tillage erosion disrupts soil structure..." and "The drive for greater operational efficiency in crop production has led to machinery innovations that have accelerated the pace of tillage erosion...", the relationship between machinery innovation and the previous content is not explained, resulting in a logical break.

Authors' response: As this study is cross-disciplinary we had to address the state-ofart provided by different disciplines - from soil science, biogeochemistry (C) to breeding science. However, we revised the entire introduction to improve the structure and internal logic for the reader. Furthermore, the new title is more specific now. It highlights the erosion aspect, addressing the focus on subsoil C and, being more precise in its character (case study).

2. Although the introduction mentions multiple research background information, it fails to clearly show how this study fills the gaps in existing research. For instance, the article points out problems such as the uncertainty in estimating root carbon input and the scarcity of carbon balance studies considering tillage erosion, but does not clarify the unique perspective and specific entry points of this study in solving these problems. As a result, the research purpose is not clear and prominent enough, which is mainly reflected in the content elaborating on the deficiencies of existing research.

Authors' response: Good point, thanks to the reviewer. We now explicitly mentioned the deficiencies and how our work can bring more information to the community.

3. The description of the background information on the soil carbon cycle and tillage erosion in the introduction part is too long, and some content has a weak direct relevance to the research question, making the overall logic less compact.

Authors' response: We completely revised the introduction and shortened this part. However, we intend to provide the state-of-the-art knowledge of underrated tillage erosion as well as the influence of erosion on the C cycle to the readers from different disciplines - to understand why we set up the specific experiment in the way we did it. For example, breeding scientists might not be aware of the spatially distributed feedbacks of soil erosion on site conditions or C sequestration potentials.

4. Although the article details the measurement methods of indicators such as root biomass, there are problems in actual operation. The minirhizotron scanning may not be able to cover all roots completely. At the same time, some unconsidered

interfering factors, such as differences in the particle composition and pore structure of different soils, may affect the morphology and distribution of roots, and thus influence the measurement of root biomass, leading to a certain degree of underestimation of the measured root biomass. In addition, when calculating the root biomass, it is assumed that the carbon content of above-ground and below-ground biomass is both 45%, and this assumption may not be consistent with the actual situation, affecting the accurate assessment of carbon input.

Authors' response: We acknowledge the limitations of minirhizotron methods. As a sampling-based approach, minirhizotrons estimate root that would grow within the soil volume occupied by the tube, following principles similar to validated methods (Fraiser, 2016). To improve accuracy, we followed established guidelines, placing tubes centrally in the planting row and scanning at 360°. We also used root metrics in volumetric units as basis for all calculations to improve comparability with other studies (Johnson, 2001). We added more information regarding that in line 165 of the revised manuscript.

A key advantage of the method is the ability to obtain repeated, non-destructive measurements over time. Image analysis allowed us to quantify changes in root length, diameter, and branching—indicators influenced by soil structure, such as compaction (Rose, 2017). While pore structure was not directly measured, these traits reflect underlying soil physical conditions.

Although the exact proportion of root biomass allocated to carbon was not measured throughout the experiment—since this would require destructive sampling—a methodological review suggests that 45% is a reasonable and widely accepted estimate for carbon input from roots (Bolinder, 2007) - This reference was added to the manuscript (see line 170). This value has been used in recent studies assessing carbon balance in agricultural systems (Heinemann, 2023).

5. The figures suffer from issues such as excessive information and lack of clarity. The axis labels, legends, etc. are not clear and explicit enough, making it difficult for readers to accurately understand the data and trends. Moreover, the labeling of some data points and the representation of error bars in the figures are not standardized, reducing the readability and credibility of the data.

Authors' response: We appreciate this observation. All figures were completely revised and presented as new version in the revised manuscript.

6. When analyzing the impacts of different soil types and wheat cultivars on biomass and carbon input, only descriptive statistical analysis is carried out, lacking in - depth exploration of the internal mechanisms. For the correlations between some results, no further causal analysis is conducted, making it difficult to reveal the underlying biological processes and ecological mechanisms.

Authors' response: We appreciate the reviewer's comment. The experimental design was specifically intended to evaluate how an erosion—deposition gradient influences the temporal development of root systems and aboveground biomass in different wheat cultivars. The goal was not to investigate underlying biological or ecological mechanisms in depth. Based on this observation, we have revised the manuscript to clearly present it as a **case study** focused on describing biomass responses to soil redistribution processes. We agree that exploring causal relationships would require a different experimental approach and more targeted data collection, and we highlight this as a recommendation for future research. Please see line 405 in the revised manuscript.

7. In the experiment, the soil moisture was measured, but the monitoring of soil nutrient dynamics was relatively scarce. The availability of soil nutrients directly affects plant growth and biomass allocation, and thus influences soil carbon input. At the same time, when simulating the impact of soil moisture on root biomass, some complex soil and plant physiological processes may be simplified, affecting the accuracy of the simulation results.

Authors' response: We appreciate the comment. We would like to emphasize that the machine learning algorithm learns patterns based on the input data it is trained on. Therefore, the broader and more representative the dataset, the better the model can capture complex interactions. In this regard, we consider our study particularly valuable, as it is a field experiment evaluating both above- and belowground biomass production of different wheat cultivars across the full erosion—deposition gradient. To the best of our knowledge, no previous study has tracked root biomass dynamics (formation and decay) throughout the entire crop season. This is especially valid for training the algorithm and correlation between above and belowground measurements. Although soil nutrient dynamics were not monitored continuously, the comprehensive data collection allows the model to indirectly capture the effects of nutrient distribution on root development.

8. Root exudates, as an important medium for the interaction between plants and soil microorganisms, have a significant impact on soil carbon input and soil health. However, the experimental design of this article does not measure root exudates, which may lead to an underestimation of root carbon input.

Authors' response: Authors' response: As the focus of this study is on root biomass production, rhizodeposition (e.g., exudates) was not considered. We have corrected this information in line 170 of the revised manuscript.

9. When discussing the research results, only partial consistencies with previous studies are briefly mentioned, and the similarities and differences between this study and other similar studies are not systematically compared. For example, when analyzing the impact of soil erosion on wheat biomass, the differences between the results of this study and those of other studies under the same or similar conditions are not compared in detail, making it difficult to highlight the value of this study.

Authors' response: We agree that our manuscript did not previously present a detailed comparison with similar studies. To address this, we have added an entire new paragraph (c.f. line 315 and 365) in the revised manuscript.

10. The discussion part has relatively few discussions on the limitations of the research, and does not fully explain the deficiencies of the research and their impacts on the results. For example, this study was only conducted at one experimental site with a limited sample size. Future research should increase the sample size and experimental sites to verify the universality of the results.

Authors' response: We appreciate the reviewer's insightful observation. In response, we have included in the discussion, methodological limitations and advantages associated with the use of minirhizotrons (see line 315 and 355 of the revised manuscript). Additionally, we now address the limitations related to the representativeness of the study area. As the study was conducted at a single field

using only three wheat cultivars, we acknowledge that this constrains the generalizability of our findings. To address this, we now emphasize the need for future research to expand sample sizes and include experimental sites across different regions with erosion—deposition gradients (line 405). Finally, we add "case study" to the title to clarify the explorative character of our work.